# Changes in ALBI Score and PIVKA-II within Three Months after Commencing Atezolizumab Plus Bevacizumab Treatment Affect Overall Survival in Patients with Unresectable Hepatocellular Carcinoma

**DOI:** 10.3390/cancers14246089

**Published:** 2022-12-10

**Authors:** Shinji Unome, Kenji Imai, Koji Takai, Takao Miwa, Tatsunori Hanai, Yoichi Nishigaki, Hideki Hayashi, Takahiro Kochi, Shogo Shimizu, Junji Nagano, Soichi Iritani, Atsushi Suetsugu, Masahito Shimizu

**Affiliations:** 1Department of Gastroenterology/Internal Medicine, Graduate School of Medicine, Gifu University, Gifu 501-1194, Japan; 2Gifu Municipal Hospital, Gifu 500-8513, Japan; 3Gifu Prefectural General Medical Center, Gifu 500-8717, Japan

**Keywords:** hepatocellular carcinoma, atezolizumab, bevacizumab, prognosis factor, ALBI score, PIVKA-II

## Abstract

**Simple Summary:**

Atezolizumab plus bevacizumab (Atez/Bev) treatment is now recommended as a first-line systemic treatment for unresectable hepatocellular carcinoma. In this study, we evaluated the therapeutic effects and adverse events of Atez/Bev treatment in the real world including patients with Child–Pugh B or non-viral hepatitis and those who received Atez/Bev treatment as a later-line treatment. Furthermore, we analyzed the factors affecting the overall survival among changes in the clinical indicators representing liver function and tumor-related factors within 3 months after the introduction of Atez/Bev treatment. The results of this study may be useful in determining whether to continue or modify Atez/Bev treatment at an early stage after starting this treatment.

**Abstract:**

In this study, we aimed to evaluate the efficacy and safety of atezolizumab plus bevacizumab (Atez/Bev) treatment for unresectable hepatocellular carcinoma (HCC) and to analyze the factors affecting overall survival (OS). A total of 69 patients who received Atez/Bev at our institutions for unresectable HCC were enrolled in this study. OS and progression-free survival (PFS) were estimated using the Kaplan–Meier method. Changes in clinical indicators within 3 months were defined as delta (∆) values, and the Cox proportional hazards model was used to identify which ∆ values affected OS. The median OS, PFS, objective response rate, and disease control rate were 12.5 months, 5.4 months, 23.8%, and 71.4%, respectively. During the observational period, 62 patients (92.5%) experienced AEs (hypertension (33.3%) and general fatigue), and 27 patients (47.4%) experienced grade ≥ 3 AEs (hypertension (10.1%) and anemia (7.2%)). There was a significant deterioration in the albumin-bilirubin (ALBI) score (−2.22 to −1.97; *p* < 0.001), and a reduction in PIVKA-II levels (32,458 to 11,584 mAU/mL; *p* = 0.040) within 3 months after commencing Atez/Bev. Both the worsening ∆ ALBI score (*p* = 0.005) and increasing ∆ PIVKA-II (*p* = 0.049) were significantly associated with the OS of patients.

## 1. Introduction

Hepatocellular carcinoma (HCC) is a prevalent disease worldwide, with approximately 800,000 individuals newly developing and dying from this malignancy each year [1]. HCC is difficult to detect during the early stages, and in most cases is diagnosed only after having progressed to an unresectable state [2]. Approximately 50% of patients with HCC receive systemic therapy [3]. Sorafenib was the first oral active multi-kinase inhibitor confirmed to be effective against unresectable HCC [4], and since its introduction, other multi-kinase inhibitors including lenvatinib, regorafenib, and cabozantinib have similarly been established to be efficacious [5,6,7].

In recent years, the importance of immune checkpoint inhibitors in HCC treatment has received increasing attention. Among these agents, treatment with atezolizumab, a programmed death-ligand 1 (PD-L1)-targeted antibody, administered in combination with bevacizumab (Atez/Bev), for unresectable HCC was for the first time reported to result in better overall survival (OS) and progression-free survival (PFS) outcomes than sorafenib (IMbrave150) [8]. On the basis of this favorable outcome, Atez/Bev is now recommended as a first-line systemic treatment for unresectable HCC in the recently revised guidelines issued in the United States, Europe, and Japan [9,10,11].

In the IMbrave150 trial [8], none of the participants had previously received systemic treatment and had a good liver functional reserve (the inclusion criterion was Child–Pugh A). However, in the real world, patients who receive Atez/Bev for unresectable HCC generally receive a range of other treatments including systemic therapy, and some patients have poor liver functional reserve, as seen in Japan, where HCC occurs in elderly patients with reduced hepatic functional reserve. In addition, the incidence of non-viral HCC, which may be less likely to respond to Atez/Bev, is also rapidly increasing [12,13]. Therefore, it is important to evaluate the efficacy and safety of Atez/Bev in clinical settings.

Although Atez/Bev has been established to be an effective treatment for unresectable advanced HCC, there are some patients who do not benefit from this treatment. Indeed, it has been found that only one-third of the patients who receive this treatment show an objective response [8]. In addition, Atez/Bev therapy can cause serious adverse events (AEs) [8]. Consequently, it is essential to identify the factors affecting survival or AEs when deciding whether to continue or discontinue treatment. In this regard, several biomarkers including PD-L1 expression and pre-existing immunity in baseline tumor tissue [14,15] have been identified as having potential utility in predicting the Atez/Bev response and in determining the course of treatment. However, such evaluations are complex and there is a need for more convenient and established biomarkers for use in daily clinical practice.

In this study, we evaluated the therapeutic effects and AEs of Atez/Bev in the treatment of unresectable HCC. Focusing on the factors affecting overall OS after the initiation of this treatment, we found that a deterioration in hepatic functional reserve and an elevation in the levels of protein induced by vitamin K absence-II (PIVKA-II) during the initial 3 months of treatment were associated with a poorer OS in these patients.

## 2. Materials and Methods

### 2.1. Enrolled Patients

A total of 69 patients who had received Atez/Bev for at least 3 months for unresectable HCC at our institutions (Gifu University Hospital, Gifu Municipal Hospital, and Gifu Prefectural General Medical Center) between November 2020 and March 2022 were included in this study. The study design was reviewed and approved by the Ethics Committee of Gifu University School of Medicine on 2 June 2021 (ethical protocol code: 2021–074).

### 2.2. HCC Diagnosis and Therapeutic Strategies

HCC was diagnosed on the basis of a typical hypervascular tumor stain on angiography and typical dynamic computed tomography (CT) or magnetic resonance imaging (MRI) findings of enhanced staining in the early phase and attenuation in the delayed phase [16]. Therapeutic strategies for HCC in this study were determined according to the clinical guidelines for HCC published by the Japan Society of Hepatology [16]. Atez/Bev was administered according to the standard regimen, for which all patients received intravenous atezolizumab (1200 mg) plus bevacizumab (15 mg/kg body weight) every 3 weeks [8]. An alternative treatment was considered when serious AEs, a hyper progressive disease defined as disease progression with a ≥2-fold increase in the first evaluation [17], or progressive disease (PD) for a certain period were observed.

### 2.3. Evaluation of the Efficacy and Safety of Atez/Bev

The therapeutic response of each patient was assessed using dynamic CT or MRI imaging according to the Response Evaluation Criteria in Solid Tumors [18]. OS was defined as the time from the day of commencing Atez/Bev therapy to death or the last visit. PFS was defined as the time from the commencement of Atez/Bev treatment to the observation of clinical disease progression or death. Adverse events were assessed according to the Common Terminology Criteria for Adverse Events (CTCAE), version 5.0.

### 2.4. Determination of Prognostic Factors and Statistical Analyses

Differences in the baseline characteristics within 3 months after the initiation of Atez/Bev therapy were compared using a paired-*t* test. Changes in clinical indicators representing liver function and tumor-related factors within 3 months after the introduction of the treatment were defined as delta (∆) values. The Cox proportional hazards model was used to identify which ∆ values affected the OS after the initiation of this treatment.

OS and PFS were estimated using the Kaplan–Meier method, and differences between curves were evaluated using the log-rank test. Maximally selected rank statistics were used to determine the optimal cut-off to maximize the separation of the curves in the two groups [19]. We used the ‘maxstat’ package (version 0.7-25) in R to conduct these statistical analyses. Statistical significance was set at *p* < 0.05, and all statistical analyses were performed using R (version 4.1.2; R Foundation for Statistical Computing, Vienna, Austria; http://www.R-project.org/, accessed on 26 July 2022).

## 3. Results

### 3.1. Patient Characteristics and HCC Treatment Status

The clinical characteristics of the enrolled patients (55 men with an average age of 74.4 years) immediately prior to the initiation of Atez/Bev treatment are shown in Table 1. With to the underlying liver diseases, 12, 22, 16, 12, and seven patients had hepatitis B virus, hepatitis C virus, non-alcoholic steatohepatitis, alcoholic liver disease, and other diseases, respectively, whereas with respect to liver functional reserve, 37, 24, seven, and one patient had Child–Pugh scores of 5, 6, 7, and 8, respectively.

Among the enrolled patients, 54 (78.2%) had received other treatment for HCC prior to the initiation of Atez/Bev, one (1.4%) had received combination treatment, and 31 (44.9%) had received other treatments after the Atez/Bev treatment. Details of pre-treatment, combination treatment, and post-treatment are shown in Table 2.

### 3.2. Efficacy and Safety of Atez/Bev for Patients with Unresectable HCC

The mean observational period for the enrolled patients was 7.8 ± 3.8 months. OS rates at 6 and 12 months and median OS were 77.6%, 50.7%, and 12.5 months, respectively (Figure 1a), whereas the PFS rates at 6 and 12 months and the median PFS were 46.0%, 24.2%, and 5.4 months, respectively (Figure 1b). The therapeutic effects of complete response, partial response, stable disease, and PD were observed in one, 14, 30, and 18 cases, respectively. The objective response rate (ORR) and disease control rate (DCR) were 23.8% and 71.4%, respectively. Fifteen patients had PD within 3 months, and these patients tended to have shorter survival than those who did not (*p* = 0.067, Appendix A).

Table 3 shows the AEs recorded in response to the Atez/Bev treatment. We found that 62 patients (92.5%) experienced some form of AE, the most frequent of which at any grade was hypertension (33.3%), followed by general fatigue (31.9%), proteinuria (26.1%), liver dysfunction (24.6%), and appetite loss (23.2%). AEs at Grade ≥ 3 were identified in 27 patients (47.4%), the most frequent of which was hypertension (10.1%), followed by anemia (7.2%), appetite loss (5.8%), and hemorrhage (5.8%). With respect to immune-related AEs, three patients experienced interstitial pneumonia, and one experienced myasthenia gravis and rheumatic arthritis. None of the enrolled patients experienced Grade 5 AEs.

### 3.3. Changes in Clinical Indicators 3 Months after the Induction of Atez/Bev Affecting OS

Table 4 shows the changes in clinical indicators representing liver functional reserve and tumor markers during the initial 3 months after the initiation of Atez/Bev treatment. Within 3 months after commencing treatment, we detected a significant deterioration in factors representing liver functional reserve including the Child–Pugh score, albumin-bilirubin (ALBI) score [20], serum albumin level, and the presence of ascites (*p* < 0.001). Moreover, there was a significant reduction in the levels of PIVKA-II (*p* = 0.040).

When analyzing the ∆ values, the ∆ Child–Pugh score, ∆ ALBI score, ∆ albumin, and ∆ T-Bil, all representing liver function impairment and ∆ PIVKA-II, were selected as prognostic factors in univariate analysis. We analyzed the ∆ ALBI score and ∆ PIVKA-II in multivariate analysis and identified a deterioration in the ALBI score (hazard ratio (HR): 5.477, 95% confidence interval (CI): 1.656–18.12, *p* = 0.005) and increased PIVKA-II (HR: 1.001, 95%CI: 1.000–1.003, *p* = 0.049) within 3 months after the initiation of Atez/Bev treatment as independent prognostic factors in multivariate analyses (Table 5). However, the AFP and PIVKA-II change ratios, which were defined by the AFP and PIVKA-II values at 3 months after Atez/Bev treatment divided by their values before the treatment, were not associated with OS (Appendix A). When limited to the 54 patients who did not have PD within 3 months, increased PIVKA-II (HR: 1.002; 95%CI, 1.001–1.003; *p* = 0.033) was the only independent risk factor for OS (Appendix A).

Maximally selected rank statistics revealed that the optimal cutoff values of the ∆ ALBI score and ∆ PIVKA-II were 0.376 and 672 mAU/mL, respectively (Appendix A). Patients with ∆ ALBI scores ≤ 0.376 (*p* < 0.001, Figure 2a) and ∆ PIVKA-II ≤ 672 mAU/mL (*p* = 0.007, Figure 2b) had significantly longer survival than those with ∆ ALBI scores > 0.376 and ∆ PIVKA-II > 672 mAU/mL, respectively. Furthermore, the enrolled patients were further divided into three groups based on using the two cutoff values as follows: Group 1, patients with ∆ ALBI score ≤ 0.376 and ∆ PIVKA-II ≤ 672 mAU/mL; Group 3, patients with ∆ ALBI score > 0.376 and ∆ PIVKA-II > 672 mAU/mL; and Group 2, patients with others. Patients in Group 1 had longer survival times than those in Group 2 (*p* = 0.012) and Group 3 (*p* < 0.001) (Figure 2c).

## 4. Discussion

In this study, we describe the clinical outcomes and AEs associated with Atez/Bev therapy for unresectable advanced HCC performed in a clinical setting. Results obtained from the updated IMbrave150 trial revealed median OS and PFS values of 19.2 and 6.9 months, and ORR and DCR values of 30% and 74%, respectively [21]. Compared with these observations, we recorded similar ORR (23.8%) and DCR (71.4%) values in the present study, whereas the median OS (12.5 months) and PFS (5.4 months) values were slightly inferior. These latter differences could be ascribed to the larger number of enrolled patients in our study who had Child–Pugh B, had received pretreatment that included other systemic therapies, or had non-viral hepatitis. In this regard, the findings of some studies have indicated that patients with Child–Pugh B or non-viral hepatitis and those who received Atez/Bev as a later-line treatment had poorer clinical outcomes [13,22,23]. In the present study, we found that patients with Child–Pugh B had significantly poorer survival than those with Child–Pugh A (Appendix A; *p* = 0.027), whereas there were no significant differences in OS among patients who received Atez/Bev as a first-line and later-line treatment (Appendix A; *p* = 0.472) or patients with viral and non-viral hepatitis (Appendix A; *p* = 0.178). Although this study included only a small number of patients, our findings nevertheless tended to indicate that the prognostic benefits of Atez/Bev may be diminished, at least in patients with reduced hepatic functional reserve. Further studies are needed to determine whether the expected effect can be achieved in cases of Atez/Bev post-treatment or in cases of non-viral hepatitis.

In this study, we established that the liver functional reserve, as indicated by the Child–Pugh and ALBI scores, was deteriorated significantly in those patients who received Atez/Bev. Furthermore, we identified an unfavorable change in ALBI score (∆ ALBI score) as a prognostic factor. Consequently, patients receiving treatment should be aware of the risk of reduced hepatic functional reserve such as deterioration in the Child–Pugh score, ALBI score, albumin levels, and the appearance of ascites, and that maintaining hepatic function reserve may improve patient prognosis. Moreover, we observed a significant reduction in the PIVKA-II levels within 3 months after the commencement of Atez/Bev treatment and identified increasing PIVKA-II (∆ PIVKA-II) as a poor prognostic factor in these patients. The response of alpha-fetoprotein (AFP), another HCC tumor marker, 6 weeks after initiating Atz/Bev therapy, has been reported to be a potential surrogate biomarker for prognosis in patients with HCC [24]. Moreover, the CRAFITY score, determined by C-reactive protein and AFP levels, has also been reported to be useful for predicting therapeutic outcomes in these patients [25]. In contrast, however, the utility of PIVKA-II assessment for predicting a response to Atz/Bev has hardly been previously reported [26]. Interestingly, even when limiting 54 patients to those who did not have PD within 3 months, ∆ PIVKA-II was the only independent risk factor for OS (Appendix A). In clinical practice, it is sometimes difficult to determine whether Atez/Bev treatment should be continued. For patients with deteriorating ALBI score and elevated PIVKA-II, especially those with a ∆ ALBI score >0.376 and ∆ PIVKA-II >672 mAU/mL belonging to Group 3 (Figure 2c), the prognosis is clearly poor, and a change to an alternative treatment should be considered.

The nature and severity of treatment-related AEs observed in this study differed substantially from those previously reported [8,21,22,23,27,28,29,30,31]. In contrast, we detected significant deterioration in liver functional reserve including albumin levels, Child–Pugh score, ALBI score, and the appearance of ascites within the initial 3 months of treatment. Although the findings of some studies have indicated that ALBI scores tend to decline within the first few weeks of treatment, observations in most previous studies have tended to indicate that these scores do not deteriorate in response to Atez/Bev [23,27,29,30,31]. The fact that we detected a positive correlation between the ∆ ALBI score and ∆ PIVKA-II in the present study (coefficient of correlation = 0.286, *p* = 0.034; Appendix A) would tend to imply that a deterioration in the ALBI score is associated with the progression of HCC itself, rather than with the AEs of this treatment. Clearly, in patients with a low hepatic functional reserve, Atz/Bev may promote a further deterioration of function. In addition, it is important to understand that when a tumor is not controlled by Atz/Bev, the liver functional reserve may deteriorate during the early stages of treatment.

This study did, however, have certain limitations, notably the fact that this was a retrospective study with a small sample size. Furthermore, the observational period was short and a substantial number of enrolled patients were censored at the end of this study. Additionally, the ∆ ALBI score and ∆ PIVKA-II, which were selected as independent risk factors for OS in this study, showed a modest positive correlation. This may have affected the reliability of the results of this study. Prospective studies involving a larger number of patients and a more extended observational period should be conducted in the future to overcome these limitations.

## 5. Conclusions

We observed a significant deterioration in ALBI score and a significant reduction in PIVKA-II levels within 3 months after initiating Atez/Bev therapy for unresectable HCC. Furthermore, a deterioration in the ALBI score and elevation of PIVKA-II within 3 months were both independent prognostic factors of the treatment. Evaluation of these factors may be useful in determining whether to continue or modify Atez/Bev treatment.

## Figures and Tables

**Figure 1 cancers-14-06089-f001:**
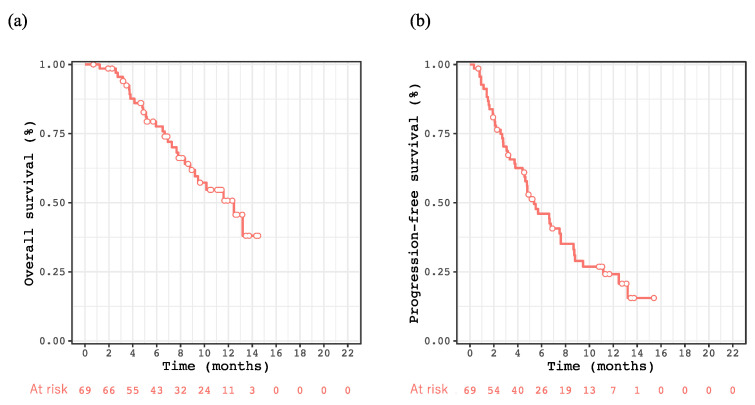
Kaplan–Meier curve for overall survival after the introduction of atezolizumab plus bevacizumab treatment for unresectable HCC (**a**) and for progression-free survival (**b**).

**Figure 2 cancers-14-06089-f002:**
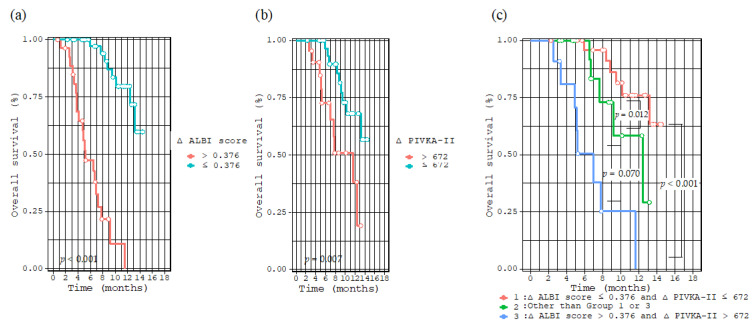
Kaplan–Meier curve for overall survival divided by ∆ ALBI score of 0.376 (**a**), ∆ PIVKA-II of 672 mAU/mL (**b**), and divided into three groups as follows: Group 1, patients with ∆ ALBI score ≤ 0.376 and ∆ PIVKA-II ≤ 672 mAU/mL; Group 3, patients with ∆ ALBI score > 0.376 and ∆ PIVKA-II > 672 mAU/mL; and Group 2, patients with others (**c**).

**Table 1 cancers-14-06089-t001:** Baseline demographic and clinical characteristics of the enrolled patients.

Variables	(*n* = 69)
Age (years)	74.4 ± 9.7
Sex (male/female)	55/14
ECOG PS (0/1/2)	55/17/2
Etiology (HBV/HCV/NASH/Alcohol/others)	12/22/16/12/7
BCLC stage (A/B1/B2/C)	9/8/16/36
Child–Pugh score (5/6/7/8)	37/24/7/1
ALBI score	−2.22 ± 0.42
ALB (g/dL)	3.5 ± 0.5
AST (U/L)	51 ± 40
ALT (U/L)	37 ± 34
T-Bil (mg/dL)	0.8 ± 0.4
PT (%)	96 ± 18
AFP (ng/mL)	2252 ± 7337
PIVKA-II (mAU/mL)	32,458 ± 156,378

Values are presented as a mean ± standard deviation. ECOG, Eastern Cooperative Oncology Group; PS, performance status; HBV, hepatitis B virus; HCV, hepatitis C virus; NASH, nonalcoholic steatohepatitis; BCLC, Barcelona Clinic Liver Cancer; ALBI score, albumin-bilirubin score; ALB, albumin; AST, aspartate aminotransferase; ALT, alanine aminotransferase; T-Bil, total bilirubin; PT, prothrombin time; AFP, alpha-fetoprotein; PIVKA-II, protein induced by vitamin K absence or antagonists-II.

**Table 2 cancers-14-06089-t002:** The pre-, combination, and post-treatment of the patients receiving atezolizumab plus bevacizumab treatment.

	Pre-Treatment	Combination Treatment	Post-Treatment
Any treatments	54 (78.2%)	1 (1.4%)	31 (44.9%)
Hepatectomy	23	0	3
RFA	23	0	2
TACE	40	1	8
Radiation therapy	9	0	2
Sorafenib	6	0	2
Regorafenib	1	0	0
Lenvatinib	18	0	21
Ramucirumab	3	0	3

RFA, radiofrequency ablation; TACE, transcatheter arterial chemo embolization.

**Table 3 cancers-14-06089-t003:** Adverse events during atezolizumab plus bevacizumab treatment.

	Any Grade (*n* = 69)	Grade 1	Grade 2	Grade ≥ 3
Any symptoms	62 (92.5%)			27 (47.4%)
Hypertension	23 (33.3%)	8 (11.6%)	8 (11.6%)	7 (10.1%)
General fatigue	22 (31.9%)	16 (23.2%)	5 (7.2%)	1 (1.4%)
Proteinuria	18 (26.1%)	5 (7.2%)	11 (15.9%)	2 (2.9%)
Liver dysfunction	17 (24.6%)	14 (20.3%)	0	3 (4.3%)
Appetite loss	16 (23.2%)	6 (8.7%)	6 (8.7%)	4 (5.8%)
Hemorrhage	11 (15.9%)	7 (10.1%)	0	4 (5.8%)
Platelet count decreased	10 (14.5%)	4 (5.8%)	3 (4.3%)	3 (4.3%)
Anemia	9 (13.0%)	2 (2.9%)	2 (2.9%)	5 (7.2%)
Diarrhea	7 (10.1%)	6 (8.7%)	1 (1.4%)	0
Hypothyroidism	5 (7.2%)	4 (5.8%)	1 (1.4%)	0
Skin disorders	2 (2.9%)	0	0	2 (2.9%)
Heart failure	2 (2.9%)	0	2 (2.9%)	0
Colonic perforation	1 (1.4%)	0	0	1 (1.4%)
Interstitial pneumonia	4 (5.8%)	3 (4.3%)	1 (1.4%)	0
Myasthenia gravis	1 (1.4%)	0	0	1 (1.4%)
Rheumatic arthritis	1 (1.4%)	0	1 (1.4%)	0

**Table 4 cancers-14-06089-t004:** Changes in clinical indicators 3 months after the introduction of atezolizumab plus bevacizumab treatment.

Variables	Introduction	After 3 Months	*p* Value
Child-Pugh score	5.6 ± 0.7	6.4 ± 1.4	<0.001
ALBI score	−2.22 ± 0.42	−1.97 ± 0.51	<0.001
ALB (g/dL)	3.5 ± 0.4	3.3 ± 0.5	<0.001
AST (U/L)	51.0 ± 40.3	49.6 ± 54.5	0.253
ALT (U/L)	37.1 ± 33.7	34.5 ± 37.3	0.364
T-Bil (mg/dL)	0.9 ± 0.4	1.3 ± 2.3	0.143
PT (%)	96.5 ± 18.3	93.1 ± 21.7	0.078
Ascites (yes/no)	0/69	15/54	<0.001
Encephalopathy (yes/no)	0/69	1/68	1.000
AFP (ng/mL)	2252 ± 7337	4997 ± 17,802	0.079
PIVKA-II (mAU/mL)	32,458 ± 156,378	11,584 ± 28,983	0.040

Values are compared using the paired-*t* test. ALBI score, albumin-bilirubin score; ALB, albumin; AST, aspartate aminotransferase; ALT, alanine aminotransferase; T-Bil, total bilirubin; PT, prothrombin time; AFP, alpha-fetoprotein; PIVKA-II, protein induced by vitamin K absence or antagonists-II.

**Table 5 cancers-14-06089-t005:** Univariate and multivariate analyses of possible risk factors for overall survival among the changes of clinical indicators within 3 months by the Cox proportional hazards model.

Variables	Univariate Analysis		Multivariate Analysis	
	HR (95%CI)	*p* Value	HR (95%CI)	*p* Value
∆ Child–Pugh score/3 months	1.971 (1.420–2.737)	<0.001		
∆ ALBI score/3 months	2.951 (1.956–4.453)	<0.001	5.477 (1.656–18.12)	0.005
∆ Albumin (g/dL)/3 months	0.167 (0.069–0.409)	<0.001		
∆ AST (U/L)/3 months	1.001 (0.991–1.011)	0.846		
∆ ALT (U/L)/3 months	0.999 (0.985–1.014)	0.925		
∆ T-Bil (mg/dL)/3 months	1.276 (1.092–1.490)	0.002		
∆ PT (%)/3 months)	0.985 (0.966–1.004)	0.120		
∆ AFP (ng/mL)/3 months	1.002 (0.999–1.004)	0.129		
∆ PIVKA-II (mAU/mL)/3 months	1.002 (1.001–1.003)	0.003	1.001 (1.000–1.003)	0.049

∆ values mean the changes of clinical indicators that represent liver function and tumor markers within 3 months after the introduction of atezolizumab plus bevacizumab treatment. ALBI score, albumin-bilirubin score; ALB, albumin; AST, aspartate aminotransferase; ALT, alanine aminotransferase; T-Bil, total bilirubin; PT, prothrombin time; AFP, alpha-fetoprotein; PIVKA-II, protein induced by vitamin K absence or antagonists-II.

## Data Availability

The data presented in this study are available upon request from the corresponding author.

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
