# Peer review of "Changes in ALBI Score and PIVKA-II within Three Months after Commencing Atezolizumab Plus Bevacizumab Treatment Affect Overall Survival in Patients with Unresectable Hepatocellular Carcinoma"

_cancers, 2022, doi:10.3390/cancers14246089_

Round 1

Reviewer 1 Report (Previous Reviewer 1)

The manuscript was revised properly and my concerns were addressed.

Reviewer 2 Report (Previous Reviewer 2)

The manuscript is revised in accordance with the reviewer's suggestions.

This manuscript is a resubmission of an earlier submission. The following is a list of the peer review reports and author responses from that submission.

Round 1

Reviewer 1 Report

This manuscript by Unome S, et al. revealed the real-world data of the prognosis of atezolizumab/bevacizumab combination therapy for advanced hepatocellular carcinoma and the factors affecting that prognosis. Although the study contains useful clinical information that worsening hepatic function and elevated PIVKA-II are associated with the poor prognosis and the modification of Atez/Bev should be considered, I think there are some concerns with the assessment and analysis of the results.

Major comments:

1. In Table 5, the univariate analysis revealed the improvement of ΔChild-Pugh score is significantly associated with poor prognosis, even though the worsening of ΔALBI score, Δalbumin, and Δbilirubin are also significant poor prognostic factors. If these results are not mistaken, this discrepancy should be discussed.

2. The chronological change of PIVKA-II is expected an exponential change, in contrast to a linear change of that of ALBI score. I think the ratio of the values is more suitable for the assessment of the change of PIVKA-II than the subtraction. (The change from 100 to 300 and from 10100 to 10300 cannot be considered to have the same meaning in the assessment of PIVKA-II.)

3. You should describe the definitions of high-risk cases and treatment strategies for these cases in detail. By setting a statistical cutoff value of ΔALBI score and ΔPIVKA-II, you should reveal how much the increase of these values worsen their prognosis. Furthermore, Kaplan-Meier analysis of these scores and the combination of that, and post-treatment of these cases may promote the understanding of these results. These additional analyses can make it easier to understand how we should treat to these cases.

4. The results of the multivariate analysis in Table 5 suggest that ΔALBI score and ΔPIVKA-II are independent poor prognostic factors, while the discussion concludes that the change in ALBI score is due to tumor progression, which is a dependent factor of PIVKA-II. This is not an accurate assessment of the results. You may reconsider the discussion.

Minor comment:

1.      "3.1. Changes in Clinical …" in the line 162 is a misnomer of “3.3.”.

Reviewer 2 Report

The authors included 69 patients of HCC treated with Atezolizumab plus Bevacizumab. They found that changes in ALBI Score and PIVKA-II within 3 months after treatment initiation were related to OS.

1.       The number of patients is too small to reach to a firm conclusion.

2.       In this study, the PFS curves drop sharply within 3 months, although patients who continued treatment for more than 3 months were included in the study. In other words, it is thought that patients who continued treatment even after PD were included in the study. In such a population, it is inappropriate to predict prognosis based on changes in data after 3 months. If such an analysis is to be done, it should be done on cases that are not PD at 3 months.